# The Novel Effector Ue943 Is Essential for Host Plant Colonization by *Ustilago esculenta*

**DOI:** 10.3390/jof9050593

**Published:** 2023-05-19

**Authors:** Shuqing Wang, Wenqiang Xia, Yani Li, Yuyan Peng, Yafen Zhang, Jintian Tang, Haifeng Cui, Lisi Qu, Tongfu Yao, Zetao Yu, Zihong Ye

**Affiliations:** 1Zhejiang Provincial Key Laboratory of Biometrology and Inspection & Quarantine, College of Life Sciences, China Jiliang University, Hangzhou 310018, China; wangsq960905@163.com (S.W.); wqxia@cjlu.edu.cn (W.X.); 13586114190@163.com (Y.L.); yuyanpeng423@163.com (Y.P.); zyfzjhzyh@163.com (Y.Z.); jintiantang@cjlu.edu.cn (J.T.); hfcui@cjlu.edu.cn (H.C.); qls1013454@163.com (L.Q.); yao_tongfu@163.com (T.Y.); yuzetaoyzt@163.com (Z.Y.); 2Institute of Crop Science, College of Agriculture and Biotechnology, Zhejiang University, Hangzhou 310012, China

**Keywords:** smut fungi, biotrophic interface, pathogen recognition, reactive oxygen species, callose

## Abstract

The smut fungus *Ustilago esculenta* obligately parasitizes *Zizania latifolia* and induces smut galls at the stem tips of host plants. Previous research identified a putative secreted protein, Ue943, which is required for the biotrophic phase of *U. esculenta* but not for the saprophytic phase. Here, we studied the role of Ue943 during the infection process. Conserved homologs of Ue943 were found in smut fungi. Ue943 can be secreted by *U. esculenta* and localized to the biotrophic interface between fungi and plants. It is required at the early stage of colonization. The *Ue943* deletion mutant caused reactive oxygen species (ROS) production and callose deposition in the host plant at 1 and 5 days post inoculation, which led to failed colonization. The virulence deficiency was restored by overexpressing gene *Ue943* or *Ue943:GFP.* Transcriptome analysis further showed a series of changes in plant hormones following ROS production when the host plant was exposed to Δ*Ue943.* We hypothesize that Ue943 might be responsible for ROS suppression or avoidance of recognition by the plant immune system. The mechanism underlying Ue943 requires further study to provide more insights into the virulence of smut fungi.

## 1. Introduction

Plant pathogenic fungi have evolved various strategies to overcome the multilayered defense system of their hosts. Among these is the secretion of small proteins that manipulate host physiology by targeting various cellular pathways, including defense responses, cell wall remodeling, signal transduction, and metabolism. Effector proteins are recognized by plant resistance proteins, triggering a cascade of defense responses known as effector-triggered immunity (ETI).

Smut fungi represent a group of biotrophic plant pathogens that cause severe diseases in crops, resulting in significant economic losses worldwide. Unlike other biotrophic pathogens, smut fungi are very suitable for effector study due to the saprophytic phase and their fully established reverse genetic manipulation system. Tens of effectors have been identified. For example, in *Ustilago maydis*, Tin2 can be utilized to stimulate the production of anthocyanins and reduce salicylic acid in local plant tissues, thus facilitating long-term colonization of the host plant [1]; Pit2 inhibits a group of apoplastic maize cysteine proteases, thereby suppressing plant defense responses [2]; Pep1 inhibits the activity of plant peroxidase enzymes involved in the production of reactive oxygen species (ROS) [3]; and See1 is necessary for *U. maydis*-induced reactivation of plant DNA synthesis during leaf tumor progression [4]. It is worth noting that Tin2 secreted by *U. maydis* and *Sporisorium reilianum* suppresses plant defense in different ways [5]. It indicates that some effectors can acquire a specialized function during coevolution with host plants [6]. Furthermore, effectors equipped by smut fungi are various and quite different from other known effectors. However, smut fungi also contain several highly conserved effectors. A core effector family is considered to contain orthologues in almost all species of smut fungi and is likely to play crucial roles in the infection process. It is interesting that core effector Pep1 was even found in the genome of some apathogenic smut fungi [7].

*U. esculenta* obligately infects *Zizania latifolia* and induces tumor formation at the late stage of infection. The genome of *U. esculenta* encodes more than 289 proteins that have signal peptides but lack transmembrane structures, making them potential candidates for virulence effectors [8]. Many putative effector genes remain functionally uncharacterized. Zhang et al. found that a secreted endoglucanase UeEgl1 can hydrolyze cellulose in the plant cell wall and promote the proliferation of mycelia in the host plant *Z. latifolia* [9].

In our previous study, time-resolved RNA-seq showed that *Ue943* is strongly expressed after mating [8]. After preliminary screening and experimental demonstration, we determined that Ue943 is one of the virulence factors required for the development of fungi in plant tissue. Homologs of *Ue943* are widely found in other smut fungi. To obtain insight into the mechanism underlying Ue943, the present research studied the localization of Ue943 and compared the plant host responses for exposure to wild-type *Ustilago* strains and mutants.

## 2. Materials and Methods

### 2.1. Strains and Growth Conditions

The *Escherichia coli* strain DH5α was utilized for cloning purposes. Compatible haploids of T-type strains UeT14 and UeT55 were isolated from germinated teliospores of smut galls induced by *U. esculenta*. The Appendix A lists the strains that were derived from UeT14 and UeT55 (Appendix A).

Haploids of *U. esculenta* were reactivated in YEPS solid medium (yeast extract 10 g/L, peptone 20 g/L, sucrose 20 g/L, agar 20 g/L) at 28 °C. Single colonies were selected and grown in YEPS liquid medium (yeast extract 10 g/L, peptone 20 g/L, sucrose 20 g/L) at 28 °C with a rotary shaker running at 200 rpm. For in vitro mating, two compatible haploid strains were mixed in equal amounts after being adjusted to an OD_600_ of 2.0, following previously established protocols [7]. The mixture was then cultured on YEPS solid medium for 60 h at 28 °C.

### 2.2. Construction of Deletion and Overexpression Strains

To knock out the target gene, homologous recombination was employed, and hygromycin B was used as a selection marker to create stable transformants [10]. Briefly, a hygromycin resistance gene was inserted between the upstream and downstream of the open reading frame, and the resulting fragment was cloned into an in-house-developed vector. Protoplasts of *U. esculenta* were generated by treating them with lywallzyme [11]. The plasmid was first linearized by restriction enzyme Sca I and then introduced into protoplasts by PEG-mediated transformation to achieve the desired gene knockout. The candidate transformants were selected on regeneration solid medium (yeast extract 10 g/L, peptone 4 g/L, sucrose 4 g/L, sorbitol 182.2 g/L, agar 15 g/L) with hygromycin B, and successful gene deletion was confirmed by PCR and RT–qPCR. Two mutated compatible haploids derived from UeT14 and UeT55 were constructed for each gene.

In addition to gene knockout, homologous recombination was also used to produce strains overexpressing the *Ue943* gene. As described before [8], the plasmid pUMa932 treated with Nco I restriction enzyme was connected with the fragment containing the promoter of the *HSP70* gene and target gene fused or not fused with a green fluorescent protein (GFP). The resulting pUMa932 recombinant plasmid was linearized by the EcoRV restriction enzyme and transformed into the *Ue943* deletion strain. The transformants were screened on a regeneration medium, and their expression was examined using confocal fluorescence microscopy or RT–qPCR. All the primers used are listed in Appendix A.

### 2.3. Quantitative PCR

Total RNA was extracted using TRIzol reagent. Tissues were first ground to a fine powder in liquid nitrogen and then extracted according to the manufacturer’s instructions. Total RNA was reverse-transcribed into cDNA using the Evo M-MLV reagent (Accurate Biology). Prior to RT–qPCR analysis, all of the cDNA samples were diluted to 50 ng/µL. Total DNA was extracted by cetyl-trimethylammonium bromide (CTAB) as previously described [12].

The primer pairs used for qPCR are listed in Appendix A. The relative expression level and the fungal biomass were evaluated using the 2^−ΔΔCt^ method. This method is based on the comparison of the cycle threshold (Ct) values of the target gene and a reference gene [13]. The expression level of the *Actin* gene in *U. esculenta* was used as a reference for gene expression quantification, and the genomic DNA of the host plant was used as a reference for fungal biomass quantification.

### 2.4. Plant Inoculation

The haploids were grown in YEPS liquid medium, resuspended in water to an OD_600_ of 2.0, and mixed at a 1:1 ratio. According to the description of Zhang et al. [14], 1 mL of the mixture was inoculated into the stem of *Z. latifolia* seedlings, followed by incubation in the greenhouse under a 12/12 h light/dark cycle at 25 ± 2 °C and 70% relative humidity. Twenty-day-old *Z. latifolia* seedlings were inoculated with two compatible haploids of *Ue943* deletion strains, wild-type (WT) strains, and overexpression gene supplement strains. The growth of mycelia in leaf sheaths and stems was monitored after 1, 3, 5, 30, or 90 days. After 90 days, the *Z. latifolia* stems were examined for macroscopic symptoms.

### 2.5. Subcellular Localization

To investigate the subcellular localization of the target protein, *U. esculenta* strains overexpressing the GFP-fused gene were constructed. Haploids were grown in YEPS liquid medium to an OD_600_ of approximately 1.0, resuspended in phosphate-buffered saline (PBS) and dropped on slides for fluorescence observation. Two compatible haploids overexpressing Ue943-GFP were inoculated into *Z. latifolia* seedlings. Infectious mycelia in leaf sheaths were observed using fluorescence microscopy 5 days post inoculation.

### 2.6. Western Blot

The *U. esculenta* haploid strain UeT55∆*Ue943-Ue943*:GFP was cultured in YEPS medium to an OD_600_ of 0.8–1.0. After centrifugation at 4000 rpm and 4 °C, the supernatant was collected, and the cell pellet was resuspended in PBS buffer. Both the cell pellet and supernatant were mixed with 4× SDS loading buffer containing DTT and boiled for 15 min to fully denature the proteins. The proteins were separated by SDS–PAGE, transferred to a polyvinylidene fluoride (PVDF) membrane, and subjected to Western blot analysis using monoclonal rabbit antibodies against epitopes of GFP sequences.

### 2.7. Microscopy Observation

The slices were imaged using a TCS-SP5 confocal microscope (Leica Microsystems, Wetzlar, Germany). Images were processed using LAS-AF software (Leica Microsystems). Callose deposition in plant tissues was stained with aniline blue. Plant slices were excited at 405 nm and observed at 440 to 460 nm. Reactive oxygen species in plant tissue were stained by 3,3′-diaminobenzene (DAB) and observed by differential interference contrast (DIC) [15,16]. Cell apoptosis was revealed by propidium iodide (PI), excited at 560 nm, and detected at 590–610 nm. Lignin deposition in the cell wall was checked by Congo red, excited at 560 nm, and detected at 590–610 nm. To visualize the mycelia of *U. esculenta* in plant tissue, FITC conjugated wheat germ agglutinin (WGA-FITC) staining was performed as described previously [17,18]. Briefly, plant tissues were fixed in Carnoy’s Fluid overnight, followed by treatment with 3.75 mM KOH at 85 °C for 60 min, and stained with 10 μg/mL WGA-FITC in PBS under vacuum. Slices were excited at 488 nm and observed at 495–530 nm. To observe the subcellular localization of the fusion protein, GFP was excited at 488 nm and the fluorescence was detected at 510 to 540 nm.

### 2.8. Transcriptome Analysis

*Z. latifolia* plants inoculated with WT strains or Δ*Ue943* strains were collected after 1, 3, and 5 days post inoculation. The integrity of the extracted total RNA was evaluated using the RNA Nano 6000 test kit on the Bioanalyzer 2100 system (Agilent Technologies, Santa Clara, CA, USA). RNA-seq was performed on an Illumina NovaSeq 6000 (Illumina, San Diego, CA, USA) following the manufacturer’s recommendations. After the quality evaluation, clean data were obtained by removing low-quality reads, reads containing sequencing adapters, and reads containing poly A or T tails. The quality of the clean data was evaluated based on the error rate, Q20, Q30, and GC (Appendix A). Clean data were mapped to the reference genome sequence of *U.esculenta* using HISAT2 (v2.2.0) with default parameters [19]. The HTseq (v0.9.1) was used to calculate the number of reads mapped to each gene, and the gene expression level was estimated by fragments per kilobase per million (FPKM) [20]. The differentially expressed genes were identified by the DESeq R package, using the Benjamini and Hochberg method to correct the obtained *p* value. Genes with adjusted *p* values < 0.05 and absolute log2 (treatment/control) values ≥ 1 were assigned as differentially expressed. The Gene Ontology (GO) annotation was performed using Blast2GO (v5.2.5), and the GO enrichment analysis was implemented using the GOseq R package with gene length bias correction [21]. We also used the Cluster Profiler R package to test the statistical enrichment of differentially expressed genes in Kyoto Encyclopedia of Genes and Genomes (KEGG) pathways [22].

### 2.9. Bioinformatic Analysis

To identify homologs of the *Ue943* gene, a Protein–Protein Basic Local Alignment Search Tool (BLASTP) search was conducted against the nonredundant protein database in NCBI (https://blast.ncbi.nlm.nih.gov/Blast.cgi, accessed on 15 January 2022). The presence of a signal peptide was predicted by SignalP 5.0 (http://www.cbs.dtu.dk/services/SignalP/, accessed on 20 January 2022). MEGA (7.0.26) was used for sequence alignment and phylogenetic analysis. The functional domain was predicted by InterPro scan (https://www.ebi.ac.uk/interpro/, accessed on 25 January 2022).

### 2.10. Statistical Analysis

The symptoms of the wild-type control and deletion mutants were compared by the chi-square test of independence. To evaluate the gene expression level, fungal biomass, ROS, and callose, we used a Student’s *t*-test or one-way ANOVA, followed by pairwise comparisons with Tukey’s post hoc test. All statistical analyses were performed with SPSS (v20.0) (SPSS Inc., Chicago, IL, USA).

## 3. Results

### 3.1. Ue943 Encodes an Important Virulence Factor in U. esculenta

The *Ue943* gene encodes a 273-amino acid protein with a predicted signal peptide (Figure 1A). Homologs of *Ue943* were found in various smut fungi, including *U. maydis*, *U. trichophora*, *U. hordei*, *S. scitamineum*, *S. reilianum, S. graminicola, Anthracocystis panici-leucophaei*, *Melanopsichium pennsylvanicum*, and others (Figure 1B, Appendix A). The sequence identity between *Ue943* and its homologs ranged from 39.53% to 64.96%. Despite its conserved homologs found in smut fungi, Ue943 lacks any known functional domain, making it challenging to elucidate its role in *U. esculenta*. Interestingly, all homologs of *Ue943* were found in smut fungi except for one that was found in *Cichorium endivia*, a dicotyledon. We suppose that it may be a result of horizontal gene transfer from the pathogen to the host plant or of contamination during sampling [23].

The expression profile during the mating and infection process was investigated by qPCR. The expression of *Ue943* was found to be strongly induced after cell fusion on YEPS solid medium or 7 days post inoculation (Figure 1C,D). Next, we evaluated the impact of *Ue943* on *U. esculenta* virulence by knocking it out (Δ*Ue943*) and then complementing it (Δ*Ue943-Ue943*). The lack of *Ue943* did not affect haploid growth or mating (Appendix A). According to macroscopic symptom development, the deletion of *Ue943* significantly reduced the virulence of *U. esculenta*, which could be restored when the *Ue943* gene was complemented (Figure 1E).

Furthermore, stem shoots of *Z. latifolia* were collected 30 days post inoculation. Fungal biomass in plant tissue was evaluated by qPCR and microscopic observation. Our results revealed that the fungal biomass of Δ*Ue943* at 30 days post inoculation was significantly lower than that of the WT strains (Figure 1F). After WGA-FICT staining, green spots signaling fungal mycelia were observed in the entire cross-section of stem shoots inoculated with WT strains, whereas no staining was observed in stem shoots inoculated with Δ*Ue943* strains (Figure 1G). These results indicate that *Ue943* plays a crucial role during the infection process.

### 3.2. Ue943 Is Secreted by U. esculenta

A bioinformatic analysis predicted a signal peptide and no transmembrane structure in Ue943. To further characterize the subcellular localization of Ue943, a GFP fusion protein was constructed and overexpressed in Δ*Ue943* strains. During the saprophytic phase of *U. esculenta*, Ue943 was found to localize at the periphery of haploid cells (Figure 2A). Western blotting showed that the Ue943:GFP fusion protein occurred in both the cell pellet and supernatant (Figure 2B and Appendix A). It is not clear how Ue943 is located at the cell periphery. Since no transmembrane domain is found in Ue943, it may be located by binding to the cell wall of *U. esculenta* or the extracellular domain of membrane proteins. On the other hand, enhanced fluorescence on the cell periphery may reflect Ue943 accumulation in secretory vesicles ready to discharge their content by exocytosis.

During the parasitic phase of *U. esculenta*, Ue943 was also localized on the periphery of the mycelia (Figure 2C). This result suggested that Ue943 might function at the biotrophic interface between the plasma membrane of fungal cells and plant cells. Additionally, we showed that the overexpression of *Ue943:GFP* effectively restored the virulence of Δ*Ue943* strains, indicating that the protein was localized to the appropriate location where it performs its function (Figure 2D).

### 3.3. ΔUe943 Mutant Induces a Hypersensitive Response

To gain insight into the function of Ue943, ROS production, cell apoptosis, and the deposition of callose and lignin were monitored during *U. esculenta* infection. Δ*Ue943* strains induced a strong ROS response around fungal mycelia at 1 day post inoculation, which disappeared by 5 days post inoculation (Figure 3A,C). No ROS production was observed during the infection with WT strains. In addition, we observed abundant callose deposition at sites where Δ*Ue943* strains attempted to penetrate the cuticle of the plant, whereas infection by WT strains did not elicit callose production (Figure 3B,D). No lignin deposition or cell apoptosis was detected during the infection of either WT or Δ*Ue943* strains (Appendix A).

To further validate these findings, we examined the expression of genes related to ROS production and callose deposition. We found that genes encoding NADPH oxidase and NADH oxidase were significantly up-regulated in plants infected by Δ*Ue943* strains at 1 day post inoculation and returned to normal levels by 5 days post inoculation (Figure 3E and Appendix A). The gene encoding callose synthase was significantly up-regulated at 5 days post inoculation (Figure 3F). Conversely, the gene encoding glucan endo-1,3-β-glucosidase, which is involved in callose degradation, was down regulated at 3 and 5 days post inoculation (Appendix A). These results are consistent with our observation of increased ROS production and callose deposition, suggesting that *Ue943* plays a critical role at the early stage of infection when parasitic mycelia attempt to penetrate the host plant cuticle.

### 3.4. Transcriptome Analysis of WT Strains and ΔUe943 Strains

*Z. latifolia* infected by WT strains or Δ*Ue943* strains was collected at 1, 3 and 5 days post inoculation and analyzed by RNA seq (Appendix A). In total, 8424 differentially expressed genes (DEGs) were identified between WT and Δ*Ue943*. Among these DEGs, 1502, 625, and 1237 genes were up-regulated, and 1265, 431, and 4037 genes were down-regulated at 1, 3, and 5 days post inoculation, respectively (Figure 4A). The Venn diagram showed that only a few genes were differentially expressed at two or more time points (Figure 4B). The expression of the DEGs was clustered and shown in a heatmap (Figure 4C).

Compared with the control (WT), the Δ*Ue943* strains induced the most significant changes 5 days post inoculation (Figure 4). KEGG enrichment analysis showed that the expression of genes related to hormone signal transduction and others was mostly changed after *Ue943* was knocked out (Figure 5). In particular, 15 out of 38 genes related to cutin, suberin, and wax biosynthesis, 24 out of 128 genes related to glutathione metabolism, 17 out of 82 genes related to ascorbate and aldarate metabolism, 58 out of 266 genes related to phenylpropane biosynthesis, and 55 out of 379 genes related to plant hormone signal transduction were differentially expressed in the WT and Δ*Ue943* strains. This is consistent with the observed ROS production and callose deposition and their impact on plant hormones (See Section 3.5). The expression of genes related to fatty acid biosynthesis and metabolism also underwent a significant change. Fatty acids are important for plasma membrane integrity and cellular homeostasis. Previous studies suggested that fatty acids can affect the activity of NADPH oxidase, which is an important source of ROS [24]. Meanwhile, ROS can oxidize polyunsaturated fatty acids (PUFAs) in cell membranes and damage the membrane structure and function [25]. In addition, GO enrichment analysis also showed intense differences in activities such as response to oxidative stress and peroxidase activity (Appendix A). The results of KEGG and GO enrichment for DEGs at 1 and 3 days post inoculation are reported in detail in Appendix A.

### 3.5. Gene Expression Changes Related to Plant Hormones

Plant hormones play a critical role in mediating the interaction between plants and pathogens. When plants are challenged by pathogens, hormones such as salicylic acid (SA), jasmonic acid (JA), ethylene (ET), abscisic acid (ABA), and auxins trigger a cascade of defense responses. Here, we found that genes related to auxin, SA, and ABA were differentially expressed in *Z. latifolia* after inoculation with the WT and Δ*Ue943* strains (Figure 6). SA participates in the defense against biotrophic pathogens by activating pathogenesis-related (PR) genes [26]. In Δ*Ue943*-infected plants, *PR1* and *PR2* were expressed at significantly higher levels, and the expression of *NPR1* was also slightly up-regulated (Figure 6A). Likewise, *PAL* (phenylalanine ammonia-lyase), which is responsible for SA biosynthesis, was up-regulated in Δ*Ue943*-infected plants.

ABA is important in the stress response of plants. Positive regulators of the ABA cascade, including ABA receptor PYL, sucrose nonfermenting 1-related protein kinases subfamily 2 (SnRK2s), and ABA insensitive protein (ABI), were up-regulated in Δ*Ue943*-infected plants (Figure 6B). Lycopene epsilon cyclase, which is involved in the biosynthesis of ABA, was also up-regulated (Figure 6B).

Auxin can promote disease symptoms in many plants. The results showed that auxin-responsive proteins, cell wall degrading enzymes, and expansins were down-regulated in Δ*Ue943*-infected plants (Figure 6C).

## 4. Discussion

Over the past few years, significant progress has been made in identifying virulence factors and understanding the role of effector proteins [27]. Smut fungi provide an excellent model for studying the mechanisms underlying plant tissue colonization by biotrophic fungi. Our previous study showed that *Ue943* encodes a putative secreted protein of unknown function. The knockout of *Ue943* leads to a significant reduction in virulence, indicating that this gene is critical for fungal infection of the host plant. In this study, we demonstrate that Ue943 can be secreted by *U. esculenta* at the biotrophic interface between the fungal and plant cells. Our results revealed that Δ*Ue943* strains induce a strong plant defense response characterized by increased ROS production and callose deposition.

Interestingly, homologs of Ue943 are only found in smut fungi and *C. endivia. C. endivia* is a dicotyledon classified in Asteraceae. Smut fungi mostly parasitize species in the grass family in an obligate biotrophic manner. A few exceptions were found in smut fungi, such as *Melanopsichium pennsylvanicum*, which colonizes the dicot genus *Persicaria* [28]. The identity between Ue943 and its homolog in *C. endivia* is 49.58%, which is higher than that in *Testicularia cyperi* (39.53%). Phylogenetic analysis showed that homologs in *C. endivia* and *Moesziomyces antarcticus* are closely related. The homolog in *C. endivia* could result from a horizontal gene transfer event or sample contamination. In any case, there may be an undiscovered smut fungus that can infect *C. endivia.*

Δ*Ue943* strains caused the production of ROS and up-regulated the expression of genes encoding NADPH and NADH oxidase in plants when *U. esculenta* tried to penetrate the epidermis (1 day post inoculation). The deposition of calloses was observed at 5 days post inoculation, following the scavenging of ROS. These results suggested that ∆*Ue943* strains were recognized by the defense system of *Z. latifolia* and induced pathogen-associated molecular pattern (PAMP)-triggered immunity [8]. RNA sequencing analysis further showed that Δ*Ue943* changed the expression of a wide range of genes related to hormone signal transduction, as well as cutin, suberin, and wax biosynthesis, glutathione metabolism, ascorbate and aldarate metabolism, and fatty acid biosynthesis and metabolism (Appendix A).

However, these results are not sufficient to elucidate the mechanism of Ue943 in the interaction between *U. esculenta* and host plants. It is not clear whether Ue943 functions by suppressing ROS production or by preventing recognition by the plant. During the saprophytic phase of *U. esculenta*, the fusion protein Ue943:GFP is localized to the periphery of haploid cells, suggesting that Ue943 may be bound to the cell wall or the plasma membrane protein of *U. esculenta*. Structure analysis of Cpl1, a homolog of Ue943 in *U. maydis,* revealed a double-Ψ-β-barrel (DPBB) architecture, which is also found in cerato-platanin proteins [29]. Cpl1 can bind to soluble chitin fragments and interact with cell-wall degrading enzymes [30]. Uvi2 is another homolog of Ue943 found in *U. hordei* [30]. Sequence identities among Ue943, Cpl1, and Uvi2 are about 60%. Cpl1 and Uvi2 showed similar structures, including the central DPBB and both grooves on the upper and lower side of the protein [30]. The deletion of Ue943 or Uvi2 strongly impairs the virulence of *U. esculenta* or *U. hordei*, but the deletion of cpl1 has only mild effects on the virulence of *U. maydis* [30,31]. Ue943 and its homologs might have divergent functions during plant infection.

Some other effectors that are located at the cell periphery are found in smut fungi. Effectors Hum3, Rsp1, and Rsp3 contain a repetitive region that is similar to repetitive repellent protein. They are tightly bound to the cell wall of smut fungi and decorate the surface of mycelium. Rsp3 can interact with secreted maize DUF26-domain family proteins, such as Afp1 and Afp2, preventing them from stimulating PAMP-triggered immunity [32,33]. On the other hand, the effector Fly1 can inactivate the chitinase secreted by the host plant, which can protect the fungal cell wall and avoid PAMP-triggered immunity induced by chitin fragments [34]. Ue943 neither contains repetitive regions nor is similar to Fly1. The function of Ue943 remains to be further studied.

Infection with Δ*Ue943* activated the SA and ABA pathways and suppressed the Auxin pathway. This occurred slightly later than ROS production and became more obvious 5 days post inoculation. As described previously, ROS can trigger the accumulation of SA, which in turn activates downstream defense genes such as PR genes [35]. SA accumulation can, in turn, inhibit ROS production [36]. ABA plays an important role in regulating stress responses. ROS and ABA act in a positive feedback loop, where ROS production stimulates ABA synthesis, and ABA accumulation leads to the production of ROS [37]. On the other hand, ABA can synergistically interact with SA to enhance the expression of PR genes [38]. It seems that ABA in *Z. latifolia* was regulated by SA but not ROS. For auxin, low content levels can enhance disease resistance in many plants because auxin weakens the plant cell wall. SA-mediated plant immunity is usually associated with the repression of the auxin signaling pathway [39]. Our transcriptome analysis exhibits a typical response in plants when they recognize a biotrophic pathogen invasion and activate the defense system.

In summary, this study provides new insights into the molecular mechanisms underlying the interaction between plants and smut fungi. The identification of Ue943 as a critical virulence factor in *U. esculenta* and its role in modulating the plant defense response and hormone signaling highlights the complex nature of the plant-fungal interaction. Further investigation into the function of Ue943 and its homologs in other fungi will deepen our understanding of the evolutionary adaptations of plant pathogenic fungi and may lead to the development of novel strategies for controlling plant diseases.

## Figures and Tables

**Figure 1 jof-09-00593-f001:**
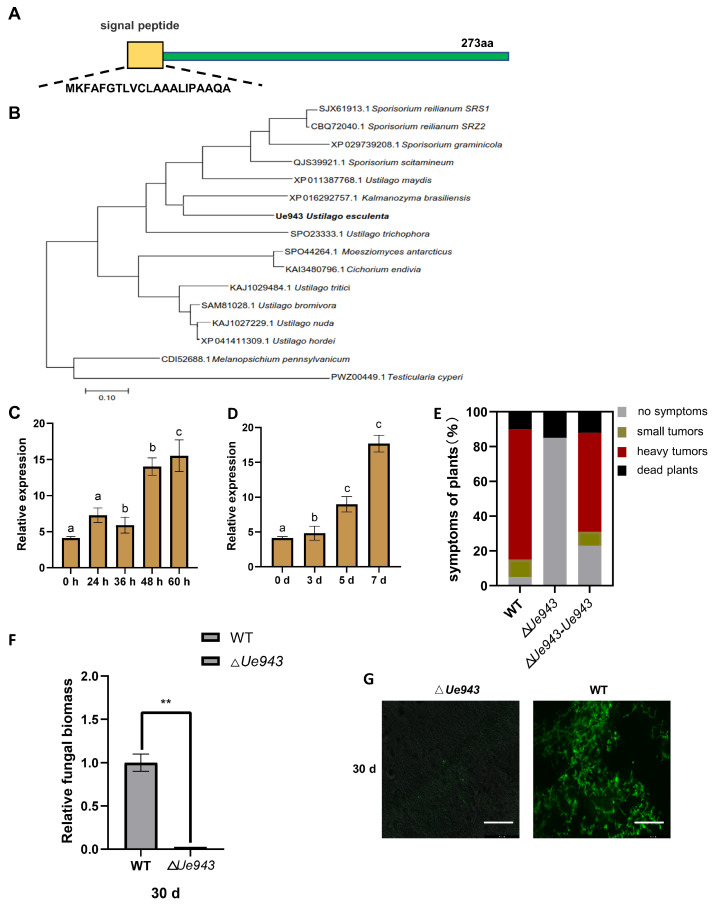
*Ue943* encodes an important virulence factor of *U. esculenta*. (**A**) Representation of Ue943 with the predicted N-terminal signal peptide. (**B**) Phylogenetic analysis of *Ue943* in *U. esculenta* and its homologs in other smut fungi. Sequences were aligned by MUSCLE. The phylogenetic tree was constructed using maximum likelihood. The name of the fungi is preceded by the accession number. (**C**) Expression profile of *Ue943* during mating on YEPS solid medium. Haploids of two compatible WT strains (UeT14 and UeT55) were mated in vitro and collected at 0 h, 24 h, 36 h, 48 h, and 60 h after culture for RT–qPCR. The data represent three independent experiments (mean ± SD). Different letters represent significant differences (*p* < 0.05). (**D**) Expression profile of *Ue943* during infection. Host plants were collected at 0, 3, 5, and 7 days after inoculation with WT strains and extracted for RT–qPCR. The data represent three independent experiments (mean ± SD). Different letters represent significant differences (*p* < 0.05). (**E**) Macroscopic symptoms were scored at 90 days post inoculation. The color code for each category is given on the left. (**F**) Fungal biomass in stem shoots evaluated with the 2^−ΔΔCt^ method at 30 days post inoculation. The genomic DNA content of *Z. latifolia* was used as a reference. The data represent three independent experiments (mean ± SD). ** *p* < 0.01. (**G**) Confocal microscopy of stem shoots from *Z. latifolia* infected by WT or Δ*Ue943*. Samples were collected 30 days post inoculation and stained with WGA-FITC (green). Scale bars represent 155.6 µm.

**Figure 2 jof-09-00593-f002:**
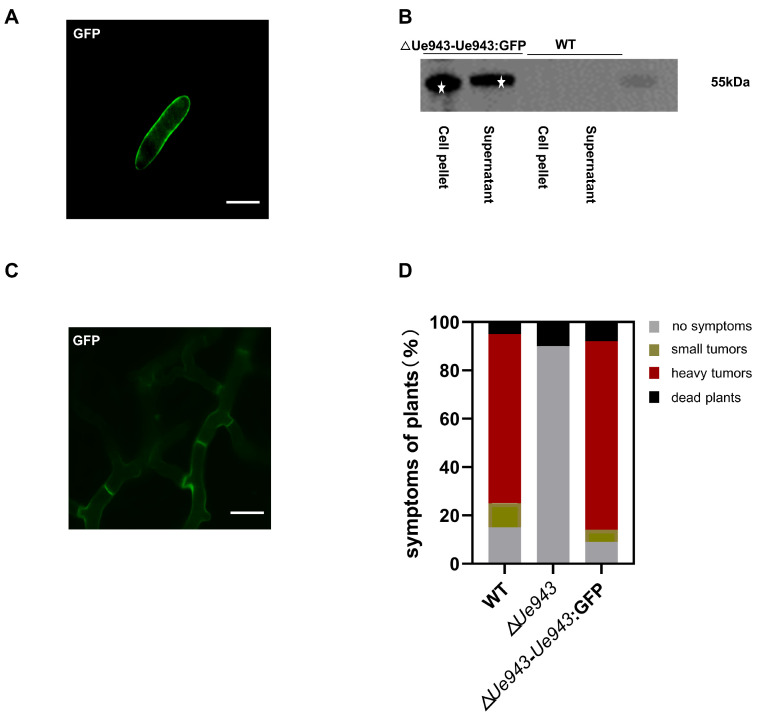
Ue943 is secreted by *U. esculenta*. (**A**) Subcellular localization of Ue943 in a haploid cell. Haploid cells overexpressing Ue943:GFP were grown in YEPS liquid medium to an OD_600_ of 1.0 and resuspended in PBS. Samples were observed using a confocal microscope. Bar, 31.25 µm. (**B**) Western blot analysis of Ue943. Ue943:GFP in the cell pellet and supernatant was analyzed using a monoclonal antibody against GFP. Stars indicate band of Ue943:GFP fusion protein. (**C**) Subcellular localization of Ue943 during infection. Leaf sheaths infected with the strains overexpressing Ue943:GFP were collected at 5 days post inoculation. Samples were observed using a confocal microscope. Bar, 15.625 µm. (**D**) Macroscopic symptoms were scored at 90 days post inoculation. The color code for each category is given on the left.

**Figure 3 jof-09-00593-f003:**
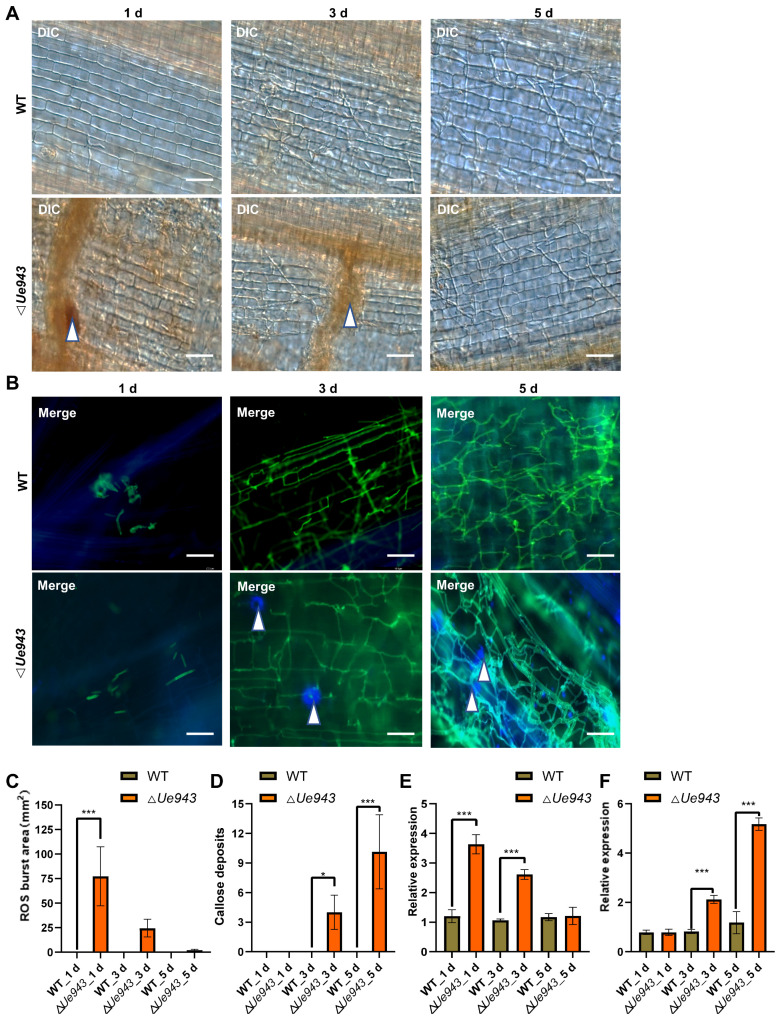
The Δ*Ue943* mutant induces plant defense response in *Zizania latifolia*. (**A**) DIC images of ROS during *U. esculenta* infection. Leaf sheaths were collected at 1, 3, and 5 days post inoculation. ROS were stained with DAB. White arrows indicate the locations where ROS were generated. Scale bar, 62.5 µm. (**B**) Fluorescence images of callose deposition during *U. esculenta* infection. Calloses were stained with aniline blue (blue), and fungal mycelia were stained with WGA-FITC (green). White arrows indicate callose deposition sites. Scale bars, 62.5 µm. (**C**) ROS production during *U. esculenta* infection. The area with ROS was analyzed by ImageJ software. The data represent three independent experiments (mean ± SD). *** *p* < 0.001. (**D**) Callose deposition during *U. esculenta* infection. A number of callose deposition spots larger than 1 mm^2^ was counted. The data represent three independent experiments (mean ± SD). * *p* < 0.05, *** *p* < 0.001. (**E**,**F**) Gene expression profile of NADPH oxidase (**E**) and callose synthase (**F**). Leaf sheaths infected by *U. esculenta* were collected at 1, 3 and 5 days post inoculation and prepared for RT–qPCR. The data represent three independent experiments (mean ± SD). *** *p* < 0.001.

**Figure 4 jof-09-00593-f004:**
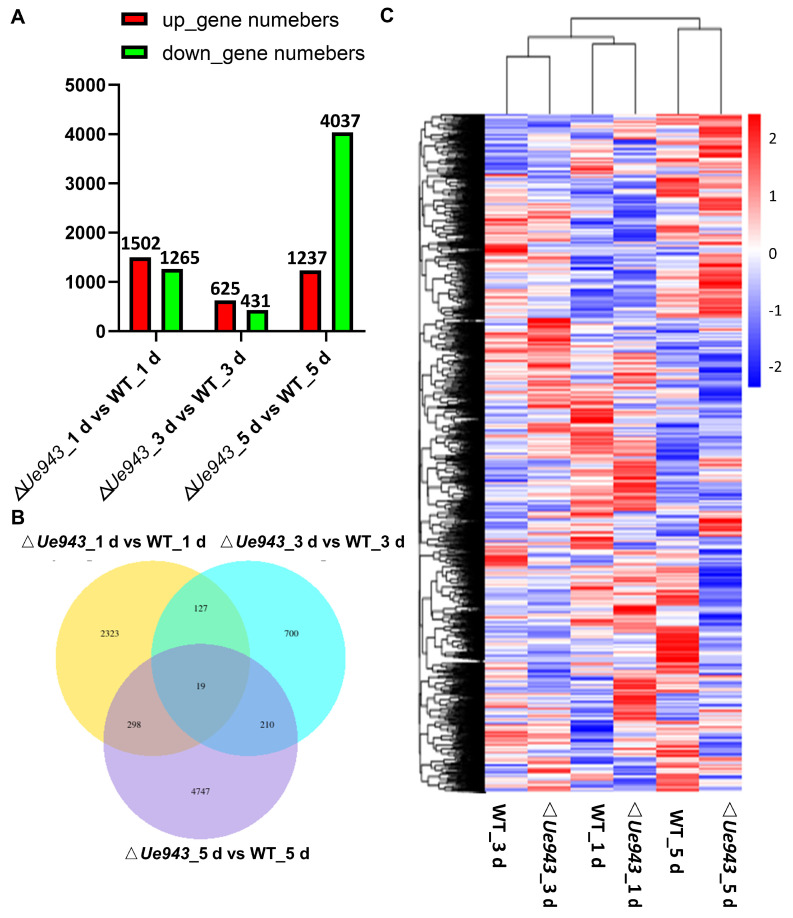
Differentially expressed genes in *Z. latifolia*. Transcriptomes of *Z. latifolia* were analyzed after inoculation with WT or Δ*Ue943* strains of *U. esculenta*. (**A**) Numbers of DEGs at 1, 3, and 5 days post inoculation. (**B**) Venn diagram showing the overlap of DEGs in *Z. latifolia* at different time points. (**C**) Hierarchical cluster analysis of DEGs in *Z. latifolia* at different time points.

**Figure 5 jof-09-00593-f005:**
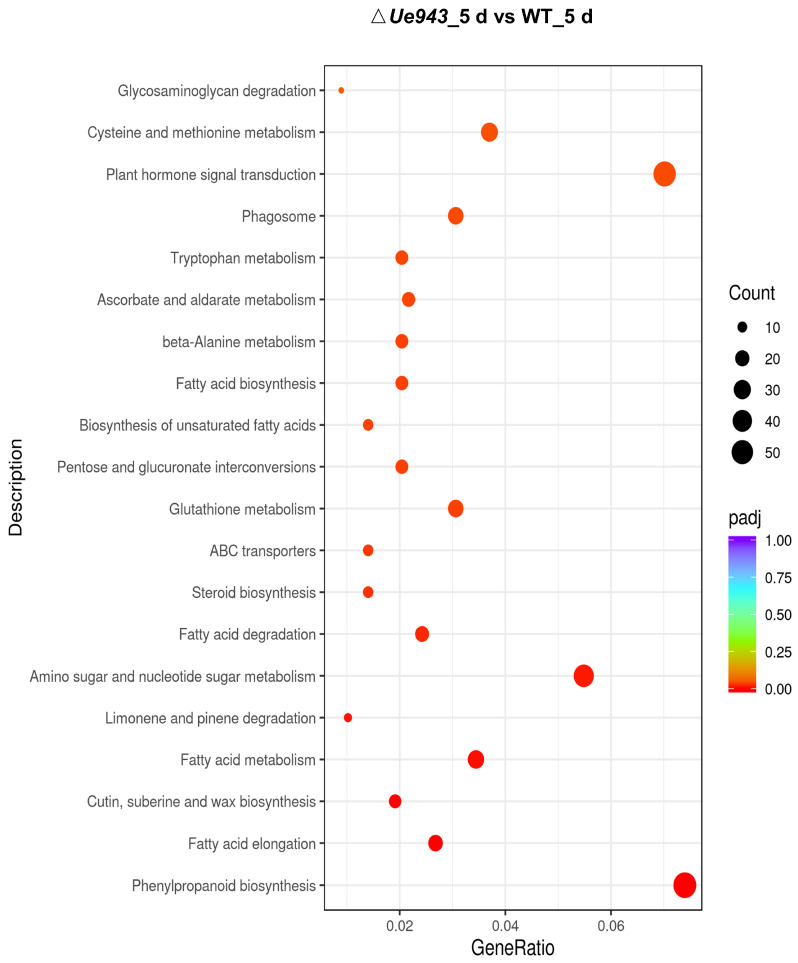
Enriched KEGG pathways for differentially expressed genes at 5 days post inoculation. Gene expression levels were compared between host plants infected by WT and Δ*Ue943* strains at 5 days post inoculation. Color represents level of significance of enrichment.

**Figure 6 jof-09-00593-f006:**
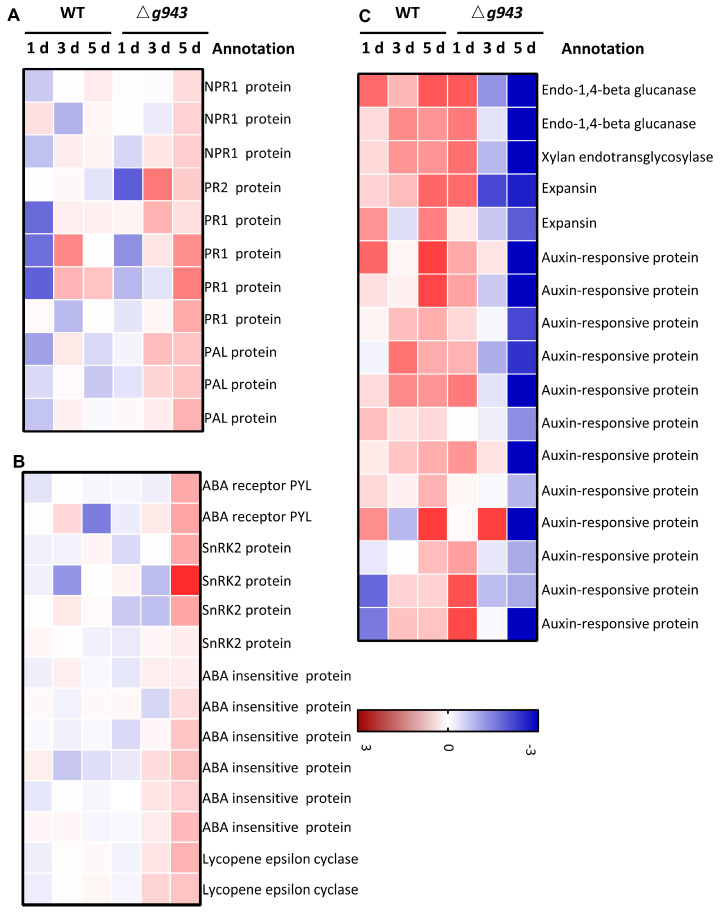
Expression profile of plant hormone related genes. (**A**) Salicylic acid-related genes; (**B**) Abscisic acid-related genes; and (**C**) Auxin-related genes general changes. These genes were differentially expressed in WT- and Δ*Ue943*-infected plants at least once.

## Data Availability

The data supporting this study’s findings are available from the author, Zihong Ye, upon reasonable request.

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
