# Peer review of "The Novel Effector Ue943 Is Essential for Host Plant Colonization by Ustilago esculenta"

_jof, 2023, doi:10.3390/jof9050593_

Round 1
Reviewer 1 Report
The manuscript submitted by Wang et al describes that the effector protein Ue943 is critical for plant colonization of the smut fungus Ustilago esculenta. Ue943 is likely conserved in smut species. The deletion mutant shows severe reduction of colonization in host plant Zizania latifolia accompanied with ROS accumulation and callose deposition. RNA-seq analysis found the upregulation of NADPH oxidase and callose synthase after plant infection, which supports the ROS production and callose deposition.
Although the function of Ue943 is still not clear in this manuscript, the work presented here seems to be carefully performed and could be a good start for future functional analysis. I suggest several major and minor points below that would help to improve the manuscript.
1) The authors used the name of effector as Ue943, which looks like just gene ID. While it would depend on the community, could the authors also name this as more standard way ? (e.g. Xyz1 etc..)
2) In page 2 line 50, I would suggest that the authors describe some examples of core effectors in smut fungi.
3) In Figure 1E and Figure 2D, the colors used in score scheme are difficult to discriminate each other (particularly between no symptoms and heavy tumors). It would be nice if the authors use other colors.
4) In page 6 line 24, the authors describe that they are not sure whether Ue943 binds to cell wall or membrane proteins. Easy experiment to demonstrate it could be to prepare protoplast of strain expressing Ue943:GFP. If cell wall protein, the fluorescence should be disappeared. If membrane protein, the fluorescence should still surround cell.
5) In Figure 3A, the authors show ROS production in infected area and the places indicated by arrows seem vascular bundles. However, for me, ROS seems to accumulate around fungal hyphae at 1 dpi in particular. If my interpretation is correct, the authors should mention this in text.
6) In page 11 line 34, “Figure 6B” comes before “Figure 6A”.
7) When I checked the orthologs of Ue943 in other smut fungi, I found the following recent report.
https://www.biorxiv.org/content/10.1101/2022.07.21.500954v1
This preprint proposes the function of the protein in Ustilago maydis. Therefore, the authors should mention it and discuss the function of Ue943 in text.
Reviewer 2 Report
The research work is very novel and the results presented are relevant for the understanding of the infectious processes of filamentous fungi. Next, I send you the observations made to your work:
Observations
Materials and methods
Line 75. It is not correct to mention "resurrected" since they were not previously dead, change the word to reactivated or any other more appropriate.
Line 88. How did you linearize the plasmid?
Results and discussion
Lines 198-199. Do you have any reference that strengthens this statement?
Check that Ue943 is in italics throughout the document.
Lines 308-351. If you are describing the results, you should not discuss them. In this paragraph, there are bibliographical references which should be part of the discussion, so they should be moved to the discussion section. Please correct.
Line 367. Is deltaUe943 correct?
Lines 372-375. A table should be included showing the genes that are affected by deltaUe943.
Reviewer 3 Report
The study investigated the mechanism of action of Ue943, a putative protein effector formerly discovered by the same research group in the smut fungus Ustilago esculenta. Combining a diversity of techniques, the study provides evidence for Ue943 participation in mitigating both early and late defence responses of host plants to the attack by the fungus. The same study also proves the widespread occurrence of Ue943 homologs in other smut fungi. The techniques employed are sound, the results obtained support the authors’ conclusion, and the paper is generally clear.
Below are reported minor observations that I invite the authors to consider.
Lines 47-49: The sentence is obscure. Tin 2 is obviously subject to natural selection. The authors should explain why the fact that Tin2 affects multiple components of the plant defence system implies that it “must evolve quickly to adapt to the host”.
Lines 61-68: the authors here provide a short summary of their results. I think the Introduction should give essential information about why and how the study was done, without anticipating the results. For example: “To get insight into the action mechanism of Ue943, the present study compared the host plant responses to exposition to wild-type Ustilago strains and mutants either lacking or overexpressing this effector………”
Line 201: “Ue943 encodes an important virulence factor of U. esculenta.” This is a conclusion supported by the data shown in the figure, not a pre-established fact. Delete
Line 213: “n: Number of infected host plants” The histograms express percentages, not absolute numbers.
Delete.
Lines 241-243: the authors suggest that Ustilago cells first secrete Ue943 in the extracellular medium and then “recruits” it at the cell periphery. This is a most illogical pathway for which the authors provide no evidence other than the occurrence of the effector at the cell periphery in cultured haploid cells (Fig 2A). In contrast, fungal hyphae in infected plant tissue show Ue943 accumulation mostly near transverse septa. In the absence of further evidence, I suggest the authors to omit inference about recruiting.
Line 251: Fig 2A indeed shows little. The cell periphery is slightly brighter than the inside, and hopefully the figure might be improves by increasing the magnification and contrast. Because of the absence of a transmembrane domain, I tend to believe that Ue943 locates in the lumen of endomembrane compartments. If so, enhanced fluorescence at the cell periphery may reflect Ue943 accumulation in secretory vesicles ready to discharge their content by exocytosis. This may permit the fungus to maintain a sufficient concentration of this and other effectors at the biotrophic interphase with the host plant, thus reducing defence responses.
Fig 2D is very similar but not identical to Fig 1E, though as far as I understand they refer to the same experiment.
Lines 248-249: “indicating that the protein was localized to the appropriate location where it performs its function (Figure 2D)”. Two problems here. First, it is not clear what the authors mean with “appropriate location” (intragenomic, cellular, extracellular or whatever). Second, the reference to Fig 2D does not seem pertinent (also see previous comment)
Line 324: what about cell periphery?
Fig. 5: Correct headline (insert space between 5d and VS and check VS)
Last, I suggest modifying the title into: The Novel Effector Ue943 is Essential for Host Plant Colonization by Ustilago esculenta.
I could find no explanation for the use of the attributive “core”, nor is this used anywhere in the main text. So, I suggest removal.
Further minor suggestions are included in the pdf attached.

I wish to congratulate the authors on their good command of English
